# Hourglass Dirac chain metal in rhenium dioxide

Shan-Shan Wang[1], Ying Liu[1], Zhi-Ming Yu[1], Xian-Lei Sheng[1,2] & Shengyuan A. Yang ᴵᴰ [1]

Nonsymmorphic symmetries, which involve fractional lattice translations, can generate exotic types of fermionic excitations in crystalline materials. Here we propose a topological phase arising from nonsymmorphic symmetries—the hourglass Dirac chain metal, and predict its realization in the rhenium dioxide. We show that $ReO_2$ features hourglass-type dispersion in the bulk electronic structure dictated by its nonsymmorphic space group. Due to time reversal and inversion symmetries, each band has an additional two-fold degeneracy, making the neck crossing-point of the hourglass four-fold degenerate. Remarkably, close to the Fermi level, the neck crossing-point traces out a Dirac chain—a chain of connected four-fold-degenerate Dirac loops—in the momentum space. The symmetry protection, the transformation under symmetry-breaking, and the associated topological surface states of the Dirac chain are revealed. Our results open the door to an unknown class of topological matters, and provide a platform to explore their intriguing physics.

---

[1] Research Laboratory for Quantum Materials, Singapore University of Technology and Design, Singapore 487372, Singapore. [2] Department of Applied Physics, Key Laboratory of Micro-nano Measurement-Manipulation and Physics (Ministry of Education), Beihang University, Beijing 100191, China. Correspondence and requests for materials should be addressed to X.-L.S. (email: xlsheng@buaa.edu.cn) or to S.A.Y. (email: shengyuan_yang@sutd.edu.sg)

Topological metals or semimetals, which host robust fermionic excitations around protected band-crossing points, have been a focus of current research. For example, Weyl and Dirac semimetals possess two- and four-fold degenerate isolated band-crossing points close to the Fermi level, around which the quasiparticles resemble the relativistic Weyl and Dirac fermions[1–16]. Under certain symmorphic symmetry operations such as mirror or inversion, the crossing points may also form one-dimensional (1D) nodal loops[17–30] or even linked nodal loops[31–34], but such loops are usually vulnerable against spin–orbit coupling (SOC) and can be removed without altering the symmetry, hence they are termed as accidental nodal loops. Recently, it was realized that nonsymmorphic symmetries, which involve fractional lattice translations, could play a key role in stabilizing the band-crossing points[35–43]. They have two important effects. First, the degeneracies enabled by nonsymmorphic symmetries could be robust against SOC. Particularly, spin–orbit nodal loops with two- or even four-fold degeneracy have been theoretically proposed[44–48]. Second, nonsymmorphic symmetries may entangle multiple bands together, so that the resulting crossing points are unavoidable and entirely dictated by the crystalline symmetry. Such band-crossing points are thus referred to as essential. For example, it was found that bands are entangled into groups of four and form hourglass-shaped dispersion on the 2D surface of nonsymmorphic insulators KHgX (X = As, Sb, Bi)[49–51]. Theoretical modeling suggested that such hourglass fermions may also exist in the bulk of 3D crystals[52], and interestingly, Bzdušek et al.[53] showed that the neck point of the hourglass may trace out a Weyl chain of two-fold-degenerate nodal loops, when multiple nonsymmorphic operations are present.

Although the essential band-crossings are solely determined by the space group for which theoretical analysis has offered valuable guidelines, the search for realistic materials that exhibit them at low energy is still challenging. This is because the bands in real materials typically have complicated 3D dispersions, such that the crossing point that we are chasing may be far away from the Fermi energy. The situation could be even worse for nodal loops, since the points on the loop are not guaranteed to have the same energy, there might be large energy variation around the loop. So far, the proposed nonsymmorphic topological metals are still limited, therefore, besides exploring new topological phases, it is also urgent to discover more suitable candidate materials to expedite experimental studies of their intriguing properties.

Here we predict a remarkable topological phase that is enabled by nonsymmorphic symmetry—the hourglass Dirac chain metal, of which the existence can be argued purely from symmetry analysis. Furthermore, based on first-principles calculations, we demonstrate that this phase is realized in an existing material—ReO₂, which serves as a solid example to illustrate the essential physics. We show that four-fold degenerate lines and hourglass-type essential band-crossings occur in the bulk band structure of ReO₂. Due to time reversal ($\mathcal{T}$) and inversion ($\mathcal{P}$) symmetries, the hourglass here is actually doubled, and the neck crossing-point here becomes a Dirac point with four-fold degeneracy. Remarkably, close to Fermi level, the neck point traces out a Dirac chain—a chain of connected (four-fold-degenerate) Dirac loops—in the momentum space, as schematically shown in Fig. 1a, b. This chain is essential, robust against SOC, and dictated by two orthogonal glide mirror planes combined with $\mathcal{T}$ and $\mathcal{P}$ symmetries. In addition, there is another pair of isolated bulk hourglass Dirac points on a symmetry line (Fig. 1a). We clarify the protection of these exotic band-crossings, and discuss their transformations under symmetry-breaking. At the sample surface, we find an interesting coexistence of drumhead-type surface states and surface Fermi arcs. The bulk hourglass Dirac chain as well as the topological surface states should be readily probed in experiment. Our findings provide an exciting platform to explore the intriguing topological fermions from nonsymmorphic symmetries.

## Results

**Symmetry and band structure**. We demonstrate the essential physics of hourglass Dirac chain using ReO₂ as a solid example. Single-crystal ReO₂ is observed with three structures denoted as $\alpha$, $\beta$, and rutile-type[54, 55]. $\beta$-ReO₂ is energetically more stable, and is experimentally shown to be a stable paramagnetic metal in a wide temperature range from the ambient temperature down to the liquid helium temperature (~4.2 K)[56, 57]. Hence we focus on $\beta$-ReO₂ here. It adopts the PbO₂-type orthorhombic crystal structure with space group no. 60 (Pbcn)[54]. As shown in Fig. 2a, the structure is characterized by zigzag chains of Re atoms running along the c-axis, and each Re atom is contained in a slightly distorted octahedron of six surrounding O atoms. As we shall see, the hourglass Dirac chain is solely dictated by the space group symmetry of the structure, which may be generated by the following symmetry operations: the inversion $\mathcal{P}$, and two glide mirror planes involving half lattice translations $\tilde{\mathcal{M}}_x : (x, y, z) \rightarrow (-x + \frac{1}{2}, y + \frac{1}{2}, z)$ and $\tilde{\mathcal{M}}_z : (x, y, z) \rightarrow (x + \frac{1}{2}, y + \frac{1}{2}, -z + \frac{1}{2})$. Here the tilde above a symbol indicates that it is a nonsymmorphic symmetry. One also notes that combining all three operations leads to a third glide mirror $\tilde{\mathcal{M}}_y : (x, y, z) \rightarrow (x, -y, z + \frac{1}{2})$. The Brillouin zone of the structure is shown in Fig. 2b.

The electronic band structure of ReO₂ is calculated by first-principles methods based on the density functional theory (DFT). SOC was included, and possible correlation effect of Re(5d) orbitals was tested. The details are presented in the Methods section. Following experimental results, we focus on the paramagnetic phase of $\beta$-ReO₂, the possibility of magnetic ordering at ultra-low temperature (<4.2 K) will be discussed later in the Discussion section. In octahedral crystal field, Re(5d) orbitals are split into $t_{2g}$ and $e_g$ groups, with the latter at higher energy. For Re⁴⁺ with 3 valence electrons, the Re-$t_{2g}$ orbitals will be half-filled, resulting in a metallic state. Figure 2c shows the calculated band structure of ReO₂ along with the projected density of states (PDOS). Indeed, one observes a metallic phase with fairly dispersive bands around Fermi level, and the low-energy states are dominated by the Re-$t_{2g}$ orbitals. Understanding that each band is at least two-fold degenerate due to the presence of $\mathcal{T}$ and $\mathcal{P}$, two interesting type of band features can be observed from Fig. 2c: Firstly, all bands are four-fold degenerate along U–X, Z–T, and T–R (Fig. 2b); secondly, hourglass-shaped

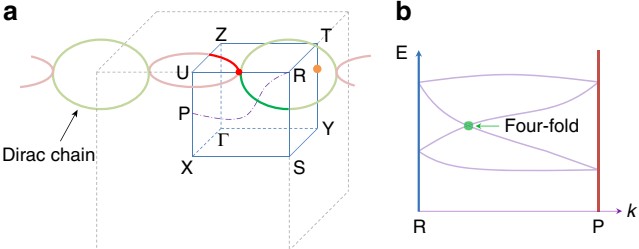

**Fig. 1** Dirac chain and hourglass dispersion. **a** Schematic figure showing Dirac chain in ReO₂, which consists of one (red) loop in $k_z = \pi$ plane and one (green) loop in $k_x = \pi$ plane. There is another isolated Dirac point (orange dot) on T–Y. These crossings are four-fold degenerate and correspond to the neck crossing-point of the hourglass-type dispersion. For example, **b** shows the schematic band dispersion along a path on the $k_x = \pi$ plane connecting R and P (an arbitrary point on U–X). Each band is two-fold degenerate, and the neck point (green dot) is four-fold degenerate

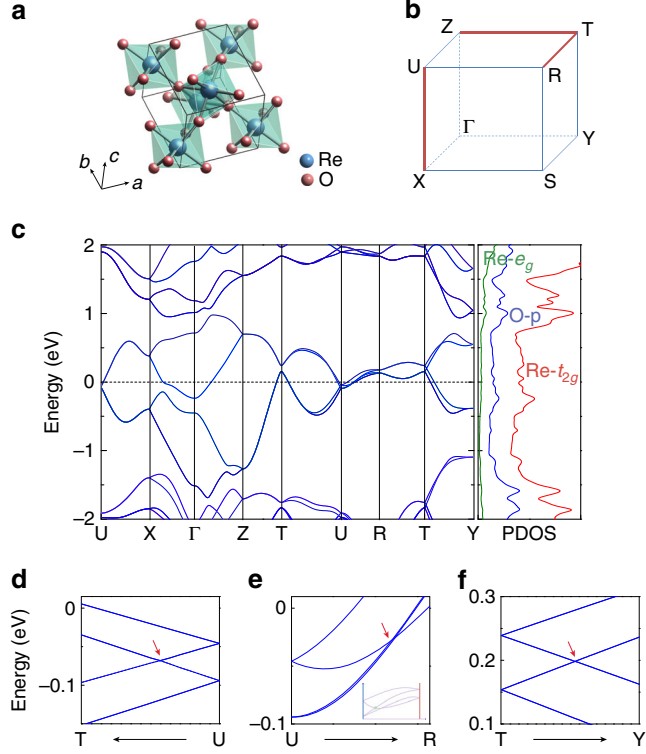

**Fig. 2** Crystal and electronic structures of ReO$_2$. **a** Crystal structure of ReO$_2$. **b** 1/8 Brillouin zone. The red lines indicate the paths where bands are four-fold degenerate. **c** Electronic band structure along with PDOS. **d–f** The enlarged band structure around the four-fold-degenerate neck-crossing points. The hourglass dispersion along U–R is somewhat distorted, as schematically shown in the inset of **e**

dispersions appear on T–U, U–R, and T–Y (Fig. 2d–f). The neck point of the hourglass is a crossing-point with four-fold degeneracy. In the following, we shall demonstrate each feature purely from the symmetry of the system.

**Four-fold degenerate high-symmetry nodal lines.** Let us first investigate the first feature regarding the four-fold degeneracy along the three high-symmetry lines. Consider the U–X line at $k_x = \pi$ and $k_y = 0$ (in unit of the inverse of respective lattice parameter). It is an invariant subspace of $\tilde{\mathcal{M}}_x$, so each Bloch state $|u\rangle$ there can be chosen as an eigenstate of $\tilde{\mathcal{M}}_x$. Since

$$\left(\tilde{\mathcal{M}}_x\right)^2 = T_{010}\overline{E} = -e^{-ik_y}, \tag{1}$$

the $\tilde{\mathcal{M}}_x$ eigenvalue $g_x$ must be $\pm i$ on U–X. Here $T_{010}$ denotes the translation along $y$ by one unit cell, and $\overline{E}$ is the $2\pi$ spin rotation. The commutation relation between $\tilde{\mathcal{M}}_x$ and $\mathcal{P}$ given by

$$\tilde{\mathcal{M}}_x\mathcal{P} = T_{110}\mathcal{P}\tilde{\mathcal{M}}_x \tag{2}$$

means that $\{\tilde{\mathcal{M}}_x, \mathcal{P}\} = 0$ on U–X. Consequently, each state $|u\rangle$ and its Kramers-degenerate partner $\mathcal{PT}|u\rangle$ must share the same $\tilde{\mathcal{M}}_x$ eigenvalue. For example, assume $|u\rangle$ has $g_x = +i$ (denoted as $|+i\rangle$), then

$$\tilde{\mathcal{M}}_x(\mathcal{PT}|+i\rangle) = -\mathcal{PT}(+i)|+i\rangle = i(\mathcal{PT}|+i\rangle), \tag{3}$$

where in the second step we used the fact that $\mathcal{T}$ is an anti-unitary operator. Same result holds for a state with $g_x = -i$. On the other hand, U–X is invariant under another anti-unitary symmetry $\tilde{\mathcal{M}}_z\mathcal{T}$, which also generates a Kramers-like degeneracy

since $\left(\tilde{\mathcal{M}}_z\mathcal{T}\right)^2 = -1$ on U–X. Note that

$$\tilde{\mathcal{M}}_x\tilde{\mathcal{M}}_z = -T_{\overline{1}00}\tilde{\mathcal{M}}_z\tilde{\mathcal{M}}_x, \tag{4}$$

where the minus sign is due to the anti-commutativity between two spin rotations, i.e., $\{\sigma_x, \sigma_z\} = 0$, so that $[\tilde{\mathcal{M}}_x, \tilde{\mathcal{M}}_z] = 0$ on U–X. Following similar derivation in Eq. (3), one finds that $|u\rangle$ and $\tilde{\mathcal{M}}_z\mathcal{T}|u\rangle$ have opposite $g_x$. Thus, the four states, $\{|u\rangle, \mathcal{PT}|u\rangle, \tilde{\mathcal{M}}_z\mathcal{T}|u\rangle, \mathcal{P}\tilde{\mathcal{M}}_z|u\rangle\}$ at the same $k$-point on U–X must be linearly independent and degenerate with the same energy. The four-fold degeneracy along Z–T and T–R can also be derived in a similar way (Supplementary Note 1).

**Hourglass dispersion and Dirac chain.** Next, we turn to the second feature regarding the hourglass dispersion. Consider the line U–R. It is invariant under both $\tilde{\mathcal{M}}_x$ and $\tilde{\mathcal{M}}_z$. From Eq. (4), $[\tilde{\mathcal{M}}_x, \tilde{\mathcal{M}}_z] = 0$ on U–R, so each state $|u\rangle$ there can be chosen as simultaneous eigenstate of both operators, with eigenvalues $(g_x, g_z) = (\pm i, \pm 1)e^{-ik_y/2}$. Using the commutation relation in (2) and $\tilde{\mathcal{M}}_z\mathcal{P} = T_{111}\mathcal{P}\tilde{\mathcal{M}}_z$, one finds that

$$\left(\tilde{\mathcal{M}}_x, \tilde{\mathcal{M}}_z\right)\mathcal{PT}|g_x, g_z\rangle = (g_x, g_z)\mathcal{PT}|g_x, g_z\rangle, \tag{5}$$

so the Kramers pair $|u\rangle$ and $\mathcal{PT}|u\rangle$ at any $k$-point on U–R share the same $(g_x, g_z)$ eigenvalues. In addition, points R and U are time-reversal invariant momenta. At R = $(\pi, \pi, \pi)$, $(g_x, g_z) = (\pm 1, \pm i)$, hence if $|u\rangle$ has eigenvalues $(g_x, g_z)$, its Kramers partner $\mathcal{T}|u\rangle$ must have $(g_x, -g_z)$. Similarly, at U = $(\pi, 0, \pi)$, since $(g_x, g_z) = (\pm i, \pm 1)$, $\mathcal{T}|u\rangle$ must have eigenvalues $(-g_x, g_z)$ if $|u\rangle$ has $(g_x, g_z)$.

Focusing on the eigenvalue $g_x$, the analysis shows that the four states in the degenerate quartet (may be chosen as $\{|u\rangle, \mathcal{T}|u\rangle, \mathcal{P}|u\rangle, \mathcal{PT}|u\rangle\}$) at R all have the same $g_x$ (+1 or −1); whereas at point U, they consist of two states with $g_x = +i$ and two other states with $g_x = -i$. Hence there has to be a switch of partners between two quartets along U–R, during which the eight bands must be entangled to form the hourglass-type dispersion. The situation is schematically shown in Fig. 3a. It is important to note that the four-fold-degenerate neck crossing-point (denoted as D on U–R) is protected because the two crossing doubly degenerate bands have opposite $g_x$ (with each degenerate pair sharing the same $g_x$, as shown in Eq. (5) and illustrated in Fig. 3a).

Furthermore, since the whole $k_x = \pi$ plane is invariant under $\tilde{\mathcal{M}}_x$, $g_x$ is well defined for any state on this plane. Hence the above argument applies to any path lying on the $k_x = \pi$ plane and connecting points U and R, which should feature an hourglass spectrum with four-fold-degenerate crossing-point in between. The crossing-point must trace out a closed Dirac loop $L_1$ on this plane, as indicated in Fig. 3b. One also notes that not only U, actually any point P on U–X has four-fold degeneracy with two $g_x = +i$ and two $g_x = -i$, as we analyzed before. Thus hourglass pattern is guaranteed to appear on any path connecting R to an arbitrary point on U–X [Fig. 3b].

Similar analysis as in the last two paragraphs applies to the $k_z = \pi$ plane, with the role played by $\tilde{\mathcal{M}}_x$ replaced by $\tilde{\mathcal{M}}_z$. It shows that hourglass pattern appears on any path connecting U to an arbitrary point on Z–T or T–R, and the neck point of the hourglass traces out a second Dirac loop $L_2$, as illustrated in Fig. 3c. Interestingly, $L_1$ and $L_2$ are orthogonal to each other, and they touch at the point D on the U–R line. Thus they constitute a Dirac chain in the momentum space, as shown in Fig. 1a.

Figure 3d, e shows the locations of the Dirac loops obtained from DFT calculations, which are consistent with our symmetry analysis. The chain is close to the Fermi level and has small energy variation (<0.2 eV). We stress that the presence of such band-crossing pattern is solely determined by the space group (plus $\mathcal{T}$). However, whether such crossings could manifest

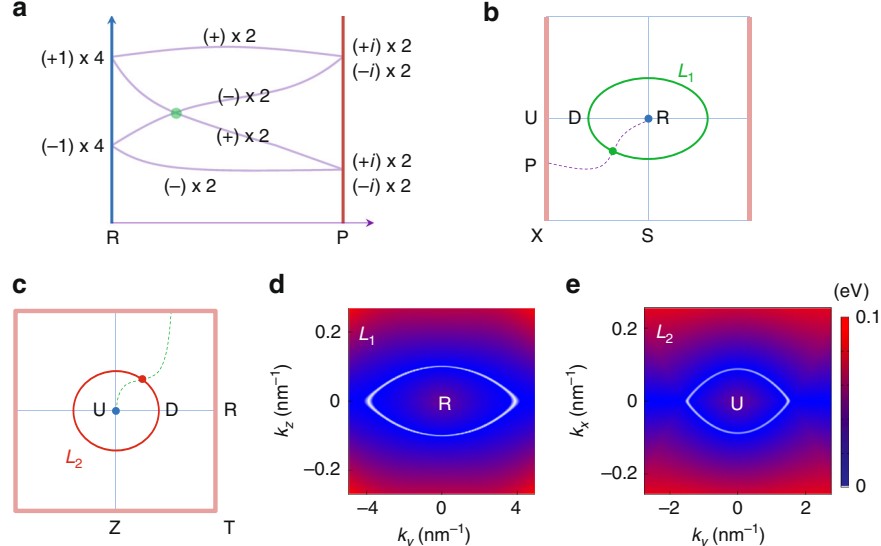

**Fig. 3** Formation mechanism of hourglass Dirac chain. **a** Schematic figure of hourglass dispersion along a path on $k_x = \pi$ plane connecting R to any point P on U–X (including U). The labels indicate the $\tilde{\mathcal{M}}_x$ eigenvalues. Partner switching between two quartets leads to the four-fold-degenerate crossing point (green dot). **b**, **c** Such crossing traces out Dirac loop **b** $L_1$ on $k_x = \pi$ plane, and also **c** $L_2$ on $k_z = \pi$ plane. The red-colored boundaries are the lines with four-fold band degeneracy. **d**, **e** Shape of two Dirac loops obtained from DFT. The color map indicates the local gap between two crossing bands

around Fermi level and have relatively small energy variation will depend on the specific material.

Up to now, one may wonder whether there exists a third loop on the $k_y = \pi$ plane, given that $\tilde{\mathcal{M}}_y$ is also a symmetry. It turns out not to be the case. Consider any state $|g_y\rangle$ on $k_y = \pi$ plane with $\tilde{\mathcal{M}}_y$ eigenvalue $g_y$, one can show that

$$\tilde{\mathcal{M}}_y\big(\mathcal{PT}|g_y\rangle\big) = -g_y\big(\mathcal{PT}|g_y\rangle\big). \tag{6}$$

Thus each Kramers pair $|u\rangle$ and $\mathcal{PT}|u\rangle$ have opposite $g_y$, which is in contrast with Eq. (5) for the other two planes. As a result, $\tilde{\mathcal{M}}_y$ can no longer protect the neck crossing-point, since each doubly degenerate band have both $\tilde{\mathcal{M}}_y$ parities and two such bands would generally hybridize to open a gap. Thus a third Dirac loop on the $k_y = \pi$ plane does not appear. This is indeed confirmed by our DFT result. Nevertheless, symmetry does dictates hourglass dispersion with an isolated Dirac point on T–Y (Fig. 2f), due to the presence of additional $\tilde{\mathcal{M}}_x$ symmetry on this line (Fig. 1a) (see Supplementary Note 2 for the analysis).

**Surface states**. Nodal loops could feature topological drumhead-like surface states[17]. We find similar phenomena for the Dirac chain here. For example, on the (001) surface, the projected loop $L_2$ is centered around $\bar{X}$ point, around which one indeed observes a pair of drumhead surface bands emanating from the projected bulk band-crossing point (Fig. 4a, b). Note that at the surface, due to the broken inversion symmetry, the spin-degeneracy of the surface bands are lifted by the strong SOC. From a slab calculation, we verify that on each surface (top or bottom), the two drumhead surface bands are indeed spin-split and non-degenerate (Supplementary Fig. 3). Similar observation is made on the (100) surface as well.

Interestingly, we find that the pair of isolated Dirac points on T–Y also generates surface Fermi arcs. As shown in Fig. 4c for the (010) surface, the arcs connect the surface-projections of the bulk Dirac points around the protection of T point in the surface Brillouin zone, similar to the Dirac semimetals Na$_3$Bi and Cd$_3$As$_2$[6, 7]. In Weyl semimetals, the surface Fermi arcs are dictated by the nontrivial topological charges (Chern number of ±1) associated with the Weyl points. However, a Dirac point

carries zero topological charge (because it consists of two Weyl points with opposite charges), hence the protection from topological charge is not guaranteed. In the current case, to reveal the possible mechanism that protects the Fermi arcs, we notice that the plane in Brillouin zone containing the points Z, T, and S is invariant under $\mathcal{T}$ and without band-crossing. Hence a 2D $\mathbb{Z}_2$ invariant can be defined for this plane and is found to be nontrivial (Supplementary Fig. 5), which dictates a Kramers pair of surface states on the T–S̃ line of the (010) surface. Thus, the Fermi arcs on this surface cannot be eliminated and is protected by the nontrivial bulk $\mathbb{Z}_2$ invariant.

**Discussion**

Our work not only reveals a hitherto unknown topological phase, it also finds an existing material for its realization. The hourglass Dirac chain revealed in ReO$_2$ represents an essential band-crossing: it is robust against SOC and dictated by the crystalline symmetry. We also studied a few other materials with the same space group symmetry, and indeed the same qualitative features can be observed in their band structures (Supplementary Note 8).

As the band-crossings discussed here are all dictated by symmetry, they must be kept as long as the space group symmetry is maintained. In Fig. 5a, b, we demonstrate that when we distort the crystal lattice while maintaining the symmetry, the shape and the size of the chain can change, but it cannot be destroyed. In contrast, if we break the symmetry, e.g. by varying the angle between $a$ and $b$ axis away from 90° (corresponding to some shear strain) to change the lattice from orthorhombic to mono-clinic, the chain will lose (part of) its protection. In this case, the distortion breaks $\tilde{\mathcal{M}}_x$ but still preserves $\tilde{\mathcal{M}}_z$ and $\mathcal{P}$, thus the Dirac loop on the $k_z = \pi$ plane is still protected (Fig. 5c), whereas the loop on the $k_x = \pi$ plane and the Dirac point on T–Y are removed. These are confirmed by the DFT calculation.

Compared with the Weyl chain proposed in ref. [53], the Dirac chain here is fundamentally different due to the doubled degen-eracy, similar to the distinction between Dirac and Weyl points. The added degeneracy comes from the preserved inversion symmetry, which is explicitly broken for the Weyl chain case. We stress that this doubling in degeneracy actually poses a more stringent condition regarding the symmetry protection of the

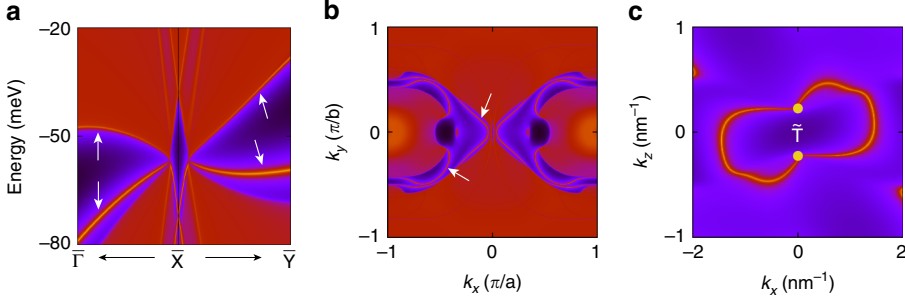

**Fig. 4** Topological surface states. **a** Projected spectrum on (001) surface, and **b** the corresponding constant energy slice at −60 meV. The arrows indicate the drumhead-like surface states. **c** Surface Fermi arcs on (010) surface connecting the surface projections of the Dirac points on T–Y (marked by the orange dots)

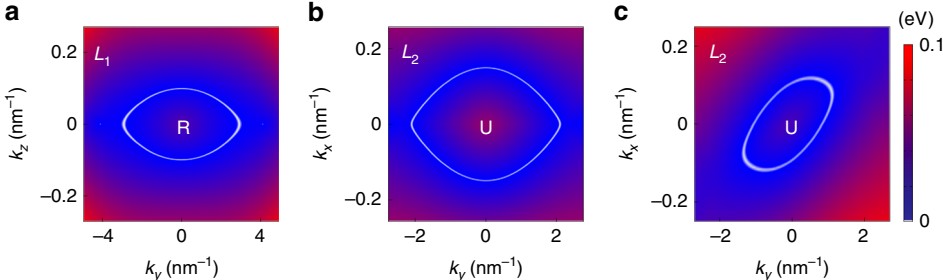

**Fig. 5** Dirac chain under lattice deformation. **a, b** Dirac chain is maintained under lattice distortion that preserves the symmetry. The figures show the two loops when lattice parameters are increased by 5%. **c** By changing the angle between a and b axis (here to 80°), symmetry is reduced and only $L_2$ loop is preserved

band-crossings: Superposing two copies of Weyl chain does not necessarily lead to a Dirac chain—they may hybridize and open a gap; the Dirac chain requires additional symmetry protection than the Weyl chain. Indeed, if there were no such degeneracy in the current case, the (missing) loop on the $k_y = \pi$ plane would be well protected. In terms of surface states, the Dirac chain metal here possesses a pair of spin-split drumhead surface bands for a single surface, although there is no spin-splitting in the bulk bands; whereas for the Weyl chain metals, both surface and bulk bands are without spin-degeneracy, and a surface typically has only one drumhead surface band. Furthermore, like the Dirac semimetal, the Dirac chain metal may also be considered as a parent phase for other topological phases (including Weyl chain metal) under symmetry breaking. For example, by suitably breaking $\mathcal{T}$ or $\mathcal{P}$, we can transform a Dirac chain metal into a Weyl chain metal or a nodal-loop metal (Supplementary Note 6).

Several recent works proposed the nodal-link semimetal phase[31–34], which also contains multiple nodal loops. However, a link structure (in which the loops do not touch each other) is topologically different from a chain. The nodal-link models in those works do not require any non-symmorphic symmetry; the linked loops are generally vulnerable against SOC; and the link is not essential in the sense that it can be removed without breaking the symmetry of the system. These features are distinct from those of the Dirac chain studied here.

In experiment, the paramagnetic metal phase of $\beta$-ReO$_2$ was shown to persist from room temperature down to the liquid helium temperature (4.2 K), and no magnetic ordering has been found[56, 57]. Hence the exotic band features reported here should be readily accessible for experimental measurements. In DFT + $U$ calculations, we find that a relatively large $U$ value could drive the system towards an antiferromagnetic (AFM) insulator phase (Supplementary Note 7), indicating the possibility of magnetic ordering at very low temperatures (at least < 4.2 K according to

experiment). Interestingly, we find that even in the AFM state (if it indeed exists), the Dirac chain or one of the Dirac loops may still be preserved if the magnetic moment is aligned along certain high-symmetry directions (Supplementary Fig. 10).

The Dirac chain, the hourglass dispersion, and the surface states are close to the Fermi level. They could be directly imaged in ARPES measurement and compared with our calculation results. Besides ARPES, we also suggest several interesting effects derived from the nontrivial bulk and surface states that could be useful for characterizing Dirac chain metals.

As for the bulk states, it was predicted that under an external magnetic field parallel to the Dirac loop plane, there will appear an almost flat Landau band at the loop energy[58]. This will lead to a pronounced peak in the density of states which can be detected by the scanning tunneling spectroscopy. For a Dirac chain as in Fig. 1a, one expects that the peak will be most pronounced when the $B$ field is along the $y$-direction (parallel to both loops), and it will be relatively small when the field is not parallel to either loop. In addition, it has been shown that the nodal-loop dispersion leads to distinct scaling in optical absorption that Im$\varepsilon(\omega)$ scales as $1/\omega$, where $\varepsilon$ is the dielectric function and $\omega$ is the light frequency[59].

The drumhead-type surface states may also lead to several interesting effects. It has been argued that they could produce a huge surface density of states, which may offer a route toward high-temperature superconductivity[60]. The recent work by Li et al.[26] attributed the unusually high surface density of states on the Be (0001) surface to the drumhead surface states, which combined with the strong electron–phonon coupling found on that surface[61] may lead to a surface superconductivity (yet to be confirmed by experiment). Interestingly, the giant enhancement of the Friedel oscillation on the Be (0001) surface was also found to be due to these nontrivial surface states[26]. In addition, with electron–electron interaction, the drumhead surface states may

lead to a surface ferromagnetism, as discussed by Liu and Balents[62]. These effects are also expected for the Dirac chain metals, and they can be detected by surface-sensitive probes such as scanning tunneling microscopy/spectroscopy (for Friedel oscillation and superconductivity) and surface magneto-optic Kerr effect (for surface magnetism). More interestingly, the orthogonal loops dictate the presence of drumhead surface states on multiple surfaces. For the case in Fig. 1a, the drumhead surface states would appear on (100) and (001) surfaces but not on the (010) surface. Thus, the different surfaces of a Dirac chain material could exhibit very different behaviors, e.g. in terms of the Friedel oscillation strength and the possible surface super-conducitivity/ferromagnetism, as determined by the surface orientation relative to the chain.

## Methods

**First-principles calculation.** The first-principles DFT calculations are performed by using the Vienna Ab-initio Simulation Package[63, 64]. The projector augmented wave (PAW) method[65] was employed to model the ionic potentials, and the generalized gradient approximation (GGA) with Perdew–Burke–Ernzerhof (PBE)[66] realization was adopted for the exchange-correlation functional. The energy cutoff was set as 400 eV. Energy and force convergence criteria are set to be $10^{-6}$ eV and 0.01 eV Å$^{-1}$, respectively. $\Gamma$-centered $k$-mesh with size $11 \times 11 \times 11$ was used for the Brillouin zone sampling. The surface states are studied using the method with maximally localized Wannier functions[67–69]. As Re(5d) orbitals may have correlation effects, we also validate our result by using the GGA + $U$ method[70]. Several on-site Hubbard $U$ values from 0 to 1.5 eV were tested, which yield no appreciable difference. Hence in the main text, we focus on the GGA results. The experimental values of the lattice parameters ($a = 4.809$ Å, $b = 5.643$ Å, $c = 4.601$ Å)[54] were used in the calculation.

**Data availability**. The data that support the findings of this study are available from the corresponding authors upon reasonable request.

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

## Acknowledgements

We thank D.L. Deng for valuable discussion. This work is supported by Singapore Ministry of Education Academic Research Fund Tier 2 (MOE2015-T2-2-144) and Tier 1 (SUTD-T1-2015004). We acknowledge computational support from the Texas Advanced Computing Center and from the National Supercomputing Centre Singapore.

## Author contributions

S.-S.W. initiated the work. S.-S.W. and X.-L.S. performed the first-principles calculations. Y.L., S.-S.W., Z.-M.Y. and S.A.Y. conducted the symmetry analysis. All authors contributed to the analysis of the results. X.-L.S. and S.A.Y. supervised the project.

## Additional information

**Competing interests:** The authors declare no competing financial interests.

