## [Peer Review File · Nature Communications]

Reviewers' comments:

Reviewer #1 (Remarks to the Author):

The authors reported on the first-principles prediction of the hourglass Dirac chain metal in ReO₂ in the combination with symmetry analyses. They defined the hourglass Dirac chain as the neck crossing point traces out a Dirac chain which connected four-fold-generated Dirac loops in the momentum space. The underlying mechanism has been in-depth discussed and the calculations have been discussed. It has been turned out to be true that the hourglass Dirac chain is protected by symmetry, rather than the electronic strongly correlated effects. Therefore, the results mainly discussed in the current manuscript using the conventional DFT calculations are reasonable and robust, if ReO₂ is a nonmagnetic. I would not like to recommend its publication in the current manuscript. However, I am asking the authors to, carefully and substantially, clarify several important aspects (below) in their revisions.

1) In ReO₂, the authors represented that, for Re⁴⁺ with 3 valence electrons, the Re-*t*_{2g} orbitals will be half-filled, resulting in a metallic state. I am wondering if the authors performed the corresponding spin-polarized DFT calculations for this current case. If ReO₂ have a half-filled Re⁴⁺ valence state, the situation of its spin configuration will split Re-*t*_{2g} orbitals in the polarized state, thereby resulting in the possible magnetism. This point will be extremely important for the current work, and if so, the band electronic structure will be altered significantly, with a possibility to form Mott-type insulator (rather than metallic). Although the early experiment demonstrated that it is paramagnetic metal (disordered spin with effective magnetic moment) at ambient condition, there was no experimental measurements as to whether or not it is nonmagnetic at a lower temperature. From the point of computational view within the DFT framework, therefore, I would like to ask the authors to doubly check this point. In general, the 5d-Re atom would not like to host magnetism. However, in most recent works, some compounds, such as Ir-containing ones (i.e., Sr-Ir-O system), was demonstrated to be magnetic. If this is the current case, it will have chance to result in the breakdown of the current conclusion in this manuscript (in particular, in the case of GGA+U cases). Could the authors please so kind to doubly check this point, at least by giving some tests.

2) The authors emphasized that these neck-connecting Dirac chains appears due to the protections of both reversal and inversion symmetries in ReO₂. Therefore, the SOC effect would not affect its occurrence. I believe that this SOC effect surely does not affect its electronic band structures in the bulk phase of ReO₂. However, the weakness of the current manuscript is the derived surface states. The authors did not specify whether or not their calculations of the non-trivial surface states include the SOC effects. Therefore, there is no doubt that, for any specified surfaces of ReO₂, the SOC effect will be highly obvious because 1) the symmetries are now broken and 2) the Re atom has the large SOC strength. Apparently, it will result in the spin-splitting. For this aspect, the authors did not have any discussions.

3) The theoretical modeling studies found that the neck connection of two Dirac chains, as shown in Fig. 1(c) in the Arxiv: 1704.00655. The other several theoretical works also discuss the related situation (Arxiv.: 1703.10886; 1704.00655, 1704.01948, 1704.0494, and so on). However, the strength of the current manuscript is providing the specified compound of ReO₂. All these works was earlier than the current manuscript. Therefore, I would like to suggest the authors to mention these works in their manuscript in a reasonable and proper place. Some related discussions should be given, in particular, to their same and difference.

4) In the discussion part, the authors mentioned that the drumhead-type non-trivial surface states could lead to huge surface density of states, which may provide a route towards high-temperature

superconductivity. Firstly, I would like to suggest that the authors show a comparison between the bulk density of states and the surface density of states of a given surface including the drumhead-type surface states induced by Hourglass Dirac Chains. Secondly, the work of pure rare-earth metal, beryllium (Be) recently mentioned that the very similar drumhead-type surface states induced by the Dirac nodal lines in its bulk phase, resulting an usually high surface density of states on the (0001) surface (Phys. Rev. Lett., 117, 096401, 2017 in Ref. 25 of this manuscript). In the early studies, the (0001) surface of the beryllium was demonstrated to have a high electron-phonon (e-ph) coupling factor as high as about 1. Therefore, it was thought to be a surface superconductivity, which was yet confirmed by experiment. I believe that it deserves to say a few more sentence to compare this case.

5) In addition, another highly charming point is the giant Friedel oscillation which would be related with its drumhead-type surface states of Dirac nodal lines as discussed in PRL, 117, 096401 (2017) and PRB, 95, 075426 (2017). The authors may add a few sentences for this point as to how the Hourglass Dirac chain behaviors on this.

In summary, through first-principles calculations with the symmetry analysis, the authors reported their prediction of the Hourglass Dirac chain metal in ReO₂ with a topological demonstration and its non-trivial drumhead surface states. However, I believe that all these results should be on basis of the prerequisite of the non-magnetic ReO₂. Although the early experimental results uncovered that ReO₂ is a paramagnetic metal only at ambient condition, the spin-polarized DFT (even DFT-U) calculations will be highly important to argue this point for the current manuscript at the extremely low temperatures. Therefore, I would like to ask the authors to doubly check this point, at least via some technical viable and reasonable tests. If the nonmagnetic feature of ReO₂ was theoretically and correctly demonstrated, I will strongly recommend its publication in Nature Communications because its predicted realization in specified materials for this kind of connected Dirac chain fermions.

Reviewer #2 (Remarks to the Author):

The manuscript by Wang et al proposed a new topological semimetal phase (the hourglass Dirac chain metal) and identified ReO₂ as a material candidate. The theoretical analyses appear to be correct and robust. I also like the fact that the proposed material candidate ReO₂ is a real compound that has been grown. For these reasons, I believe the paper is certainly valuable.

On the other hand, I also think that the so-called "new" topological phase here is quite similar to the Weyl nodal chains proposed in Ref. 47. As the authors admitted, the only difference is that the degeneracy of the bands. Thus it is highly desirable that the authors can better elaborate on why the proposed Dirac chain metal is different from the Weyl chain metals in Ref. 47 in terms of the bulk states, the surface states. Also, considering the fact that there have been many "new" topological semimetal phases proposed in the last 2-3 years, the authors should better explain how the proposed Dirac nodal chain metals can lead to unique possible transport and optical effects that are distinct from previously proposed topological semimetal phases.

Reviewer #1

General Comment: *“The authors reported on the first-principles prediction of the hourglass Dirac chain metal in ReO₂ in the combination with symmetry analyses. They defined the hourglass Dirac chain as the neck crossing point traces out a Dirac chain which connected four-fold-generate Dirac loops in the momentum space. The underlying mechanism has been in-depth discussed and the calculations have been discussed. It has been turned out to be true that the hourglass Dirac chain is protected by symmetry, rather than the electronic strongly correlated effects. Therefore, the results mainly discussed in the current manuscript using the conventional DFT calculations are reasonable and robust, if ReO₂ is a nonmagnetic. I would not like to recommend its publication in the current manuscript. However, I am asking the authors to, carefully and substantially, clarify several important aspects (below) in their revisions.”*

Reply: We thank the reviewer for summarizing our work and for considering our result as “reasonable and robust”. We shall fully address the valuable comments from the reviewer in the following.

Comment 1: *“In ReO₂, the authors represented that, for Re⁴⁺ with 3 valence electrons, the Re-t_{2g} orbitals will be half-filled, resulting in a metallic state. I am wondering if the authors performed the corresponding spin-polarized DFT calculations for this current case. If ReO₂ have a half-filled Re⁴⁺ valence state, the situation of its spin configuration will split Re-t_{2g} orbitals in the polarized state, thereby resulting in the possible magnetism. This point will be extremely important for the current work, and if so, the band electronic structure will be altered significantly, with a possibility to form Mott-type insulator (rather than metallic). Although the early experiment demonstrated that it is paramagnetic metal (disordered spin with effective magnetic moment) at ambient condition, there was no experimental measurements as to whether or not it is nonmagnetic at a lower temperature. From the point of computational view within the DFT framework, therefore, I would like to ask the authors to doubly check this point. In generally, the 5d-Re atom would not like to host magnetism. However, in most recent works, some compounds, such as Ir-containing ones (i.e., Sr-Ir-O system), was demonstrated to be magnetic. If this is the current case, it will have chance to result in the breakdown of the current conclusion in this manuscript (in particular, in the case of GGA+U cases). Could the authors please so kind to doubly check this point, at least by giving some tests.”*

Reply: We thank the reviewer for the valuable suggestion. Indeed, the magnetic state of the material β -ReO₂ is crucial for our analysis. Our main results are based on that β -ReO₂ being a paramagnetic metal. We adopted the paramagnetic phase in our calculation and analysis, because this is the phase observed in experiment under

ambient condition. The reviewer raised an important question that whether the paramagnetic phase can be maintained at a lower temperature.

To address this question, first of all, we have carefully searched the literature on the low-temperature experiments on β -ReO₂. We found that in the work by Rogers et al. [*Inorganic Chemistry* **8**, 841 (1969) cited as Ref.56 in revised manuscript], the properties of β -ReO₂ have been systematically studied from room temperature down to liquid helium temperature (4.2 K). The finding is that β -ReO₂ remains a paramagnetic metal in this wide temperature range down to 4.2 K, and there is no magnetic phase transition observed. This is also consistent with another experiment by J. B. Goodenough et al. [*CR Hebd. S 'eances Acad. Sci.* **261**, 2331–2343 (1965) cited as Ref.55 in revised manuscript]. For this latter reference, we were not able to find a copy in English. So we consulted Prof. J. B. Goodenough at the University of Texas at Austin through private communication. Prof. Goodenough kindly responded us and confirmed that β -ReO₂ was a paramagnetic metal down to 4.2 K, and *no* magnetic ordering was observed. From these evidences, we can conclude that β -ReO₂ maintains the paramagnetic metal phase in a wide temperature range from room temperature to liquid helium temperature. If there exists any magnetically ordered state, it could only occur at a very low temperature below 4.2 K. We note that the wide temperature range from 4.2 K to 300 K covers the range that is mostly accessible for current experimental techniques on topological materials (e.g. ARPES measurements are typically performed around 30 K, according to our experimental colleague Prof. Tian Qian from the Institute of Physics in Beijing). It is also noted that the Weyl chain material IrF₄ discussed by Bzdušek et al. [*Nature* **538**, 75 (2016)] is only paramagnetic above a transition temperature \sim 100 K, yet still considered accessible for ARPES measurement.

Next, following the reviewer's suggestion, we have tested several different magnetic configurations (see Fig.R2) in GGA+U calculations (SOC included). We find the following results. (i) The ferromagnetic (FM) configurations are always energetically unfavorable. (ii) At $U=0$ eV, all magnetic configurations, including FM and antiferromagnetic (AFM) cases, will converge to the nonmagnetic result, with zero magnetic moments. (iii) At large $U>1.0$ eV, AFM state appears to have a lower energy. For $U=1.5$ eV, the AFM configuration in Fig.R1(e) is found to have the lowest energy, with an energy \sim 0.152 eV (per unit cell) lower than the nonmagnetic state and a magnetic moment \sim 0.76 μ_B per Re site. Its band structure in Fig.R2 shows a Mott insulating phase. Hence, from calculation, it is possible to have an AFM phase as suggested by the reviewer, but according to the experimental results, it can only occur at very low temperature at least below 4.2 K. [In view of such low transition temperature (if it exists), it is difficult to theoretically pin down the ground state close to absolute zero temperature, because the exact value of U cannot be determined and other instabilities like superconductivity may also take place.]

In addition, we find that if the AFM state indeed occurs in β -ReO₂, interestingly, the Dirac chain or loop may still be preserved when the magnetic moment is aligned along certain high-symmetry directions. This is schematically shown in the following Fig.R3 and confirmed by our calculation. The reason is that for these cases, the AFM ordering preserves the combined PT symmetry although the individual P and T are broken; and also one (or two) glide mirror is still maintained, so that our argument of the symmetry-protection still applies. Thus, the band-crossing pattern in the AFM state here can in fact be controlled by the magnetic moment alignment.

From the above discussion, the conclusions are: (i) β -ReO₂ has been demonstrated in experiment as a paramagnetic metal in a wide temperature range from room temperature down to 4.2 K, so the predicted Dirac chain is robust and readily accessible to experimental measurement; (ii) from GGA+U calculation, it is possible to have AFM state energetically favored, however, the transition temperature to the AFM state must be very low (< 4.2 K according to experiment); (iii) the Dirac chain (or loop) may still be preserved in the AFM state if the magnetic moment is aligned in certain high-symmetry directions.

Again we thank the reviewer for the valuable suggestion, which helps to make our study more complete. In the revised manuscript, we have added the following sentence on Page 5:

“ β -ReO₂ is energetically more stable, and is experimentally shown to be a stable paramagnetic metal in a wide temperature range from the ambient temperature down to the liquid helium temperature (~ 4.2 K)^{55,56}.”

And the following discussion has been added on Page 13:

“In experiment, the paramagnetic metal phase of β -ReO₂ was shown to persist from room temperature down to the liquid helium temperature (4.2 K), and no magnetic ordering has been found^{55,56}. Hence the exotic band features reported here should be readily accessible for experimental measurements. In DFT+U calculations, we find that a relatively large U value could drive the system towards an antiferromagnetic (AFM) insulator phase (see Supplementary Information), indicating the possibility of magnetic ordering at very low temperatures (at least < 4.2 K according to experiment). Interestingly, we find that even in the AFM state (if it indeed exists), the Dirac chain or one of the Dirac loops may still be preserved if the magnetic moment is aligned along certain high-symmetry directions (see Supplementary Information).”

And we have added the test results of the magnetic configurations including Fig.R1 to Fig.R3 into the Supplementary Information.

Figure R1 | Magnetic configurations tested in our calculation. We have scanned the magnetic moment orientations in the three high symmetry planes. Here (a-c) are for the ferromagnetic ordered states. (d-i) are for two types of antiferromagnetic ordered states.

Figure R2 | Calculated band structure for the AFM configuration in Fig.R1(e) with $\theta = 135^\circ$ and $U = 1.5$ eV.

Figure R3 | Schematic figure showing that the Dirac chain or loop could still be preserved in certain AFM states if the magnetic moment is along high-symmetry directions. In each sub-figure, the upper panel shows the AFM configuration, and the lower panel illustrates the preserved band-crossings. (a) If the magnetic moment is aligned with the b -axis, the Dirac chain is preserved. (b,c) If the moment is aligned with the c or a axis, one Dirac loop is preserved.

Comment 2: “The authors emphasized that these neck-connecting Dirac chains appears due to the protections of both reversal and inversion symmetries in ReO_2 . Therefore, the SOC effect would not affect its occurrence. I believe that this SOC effect surely does not affect its electronic band structures in the bulk phase of ReO_2 . However, the weakness of the current manuscript is the derived surface states. The authors did not specify whether or not their calculations of the non-trivial surface states include the SOC effects. Therefore, there is no doubt that, for any specified surfaces of ReO_2 , the SOC effect will be highly obvious because 1) the symmetries are now broken and 2) the Re atom has the large SOC strength. Apparently, it will result in the spin-splitting. For this aspect, the authors did not have any discussions.”

Reply: We thank the reviewer for the valuable comment. We confirm that SOC is included for both bulk and surface calculations. The reviewer is absolutely correct that SOC will have a big effect at surface, generally resulting in spin-splitting of the surface states. We checked that this is indeed the case here. Note that this information cannot be easily seen from the projected spectrum as in Fig.5(a) of the main text. Hence we performed a calculation of a slab. The result plotted in Fig.R4 below explicitly shows that on *each* surface there is a pair of spin-split surface bands (we have checked the state degeneracy and the wave-function distribution). In light of this result, back to Fig.5(a), we now realize that the bright line which slightly overlaps with the bulk bands is actually the other spin-split surface band [see revised Fig.5(a)].

In the revised manuscript, we have added the following discussion on Page 10:

“Note that at the surface, due to the broken inversion symmetry, the spin-degeneracy of the surface bands are lifted by the strong SOC. From a slab calculation, we verify that on each surface (top or bottom), the two drumhead surface bands are indeed spin-split and non-degenerate (see Supplementary Information).”

And Fig.R4 has been added into the Supplementary Information and its details are discussed in Supplementary Note 4.

Figure R4 | (a) shows the result of a slab with (001) surface orientation and a thickness of 200 unit cells. From the wave-function distribution, we verify that on each surface (top or bottom), there is one pair of spin-split surface bands (marked in purple color). In the obtained data, each purple line still has a double-degeneracy because it includes states from both top and bottom surfaces. To further demonstrate the point more explicitly, we apply a surface potential of -0.04 eV on the topmost layer, which is expected to shift the surface states on the top surface down in energy. This is confirmed by the calculation result plotted in (b). The top and bottom surface bands are now completely separated. One hence explicitly verifies that on each surface there is a pair of spin-split non-degenerate surface bands.

Comment 3: *“The theoretical modeling studies found that the neck connection of two Dirac chains, as shown in Fig. 1(c) in the Arxiv: 1704.00655. The other several theoretical works also discuss the related situation (Arxiv.: 1703.10886; 1704.00655, 1704.01948, 1704.0494, and so on). However, the strength of the current manuscript is providing the specified compound of ReO₂. All these works was earlier than the current manuscript. Therefore, I would like to suggest the authors to mention these works in their manuscript in a reasonable and proper place. Some related discussions should be given, in particular, to their same and difference.”*

Reply: We thank the reviewer for the suggestion. We have cited these papers in the revised manuscript (in both Introduction and Discussion sections). These mentioned

works are all focused on two (or more) nodal loops forming a link structure (in which the loops do not touch each other), which is topologically different from a chain (in which the two loops touch at a point). The mentioned result in Fig.1(c) of Arxiv:1704.00655 [recently published as Phys. Rev. B 96, 041103(R) (2017)] is a fine-tuned special case of the studied model, which does not have a symmetry/topology protection. The differences between those works and ours also include that: the nodal-link models in those works do not require any non-symmorphic symmetry; the linked loops are generally vulnerable against SOC; and the link is *not* essential in the sense that it can be removed without breaking the symmetry of the system. In comparison, the Dirac chain studied here is robust against SOC, and non-symmorphic symmetries are crucial for generating the hourglass-type dispersion, which in turn guarantees the presence of the Dirac chain.

Following the reviewer's suggestion, we have added the following discussion on Page 12 and 13:

“Several recent works proposed the nodal-link semimetal phase³⁰⁻³³, which also contains multiple nodal loops. However, a link structure (in which the loops do not touch each other) is topologically different from a chain. The nodal-link models in those works do not require any non-symmorphic symmetry; the linked loops are generally vulnerable against SOC; and the link is not essential in the sense that it can be removed without breaking the symmetry of the system. These features are distinct from those of the Dirac chain studied here.”

Comment 4: *“In the discussion part, the authors mentioned that the drumhead-type non-trivial surface states could lead to huge surface density of states, which may provide a route towards high-temperature superconductivity. Firstly, I would like to suggest that the authors show a comparison between the bulk density of states and the surface density of states of a given surface including the drumhead-type surface states induced by Hourglass Dirac Chains. Secondly, the work of pure rare-earth metal, beryllium (Be) recently mentioned that the very similar drumhead-type surface states induced by the Dirac nodal lines in its bulk phase, resulting an usually high surface density of states on the (0001) surface (Phys. Rev. Lett., 117, 096401, 2017 in Ref. 25 of this manuscript). In the early studies, the (0001) surface of the beryllium was demonstrated to have a high electron-phonon (e-ph) coupling factor as high as about 1. Therefore, it was thought to be a surface superconductivity, which was yet confirmed by experiment. I believe that it deserves to say a few more sentence to compare this case.”*

Reply: We thank the reviewer for the valuable suggestions. Following the first suggestion, we have calculated and compared the bulk and surface density of states. The result is shown in Fig.R5 below, from which one can observe the contribution from the drumhead surface states. Because of the relatively strong surface band

dispersion (which is partly due to the spin-splitting discussed above), the surface state contribution here does not form a very sharp peak in energy. We have added this figure and a discussion (Supplementary Note 4) into the Supplementary Information.

Following the second suggestion, we have also added the following discussion regarding the possible surface superconductivity of Be on Page 14:

“The recent work by Li et al.²⁵ attributed the unusually high surface density of states on the Be (0001) surface to the drumhead surface states, which combined with the strong electron-phonon coupling found on that surface⁶⁰ may lead to a surface superconductivity (yet to be confirmed by experiment).”

Figure R5 | The bulk density of states (blue curve) in comparison with the surface density of states (red curve) for the (001) surface. Their difference reflects the contribution from the drumhead-like surface states.

Comment 5: *“In addition, another highly charming point is the giant Friedel oscillation which would be related with its drumhead-type surface states of Dirac nodal lines as discussed in PRL, 117, 096401 (2017) and PRB, 95, 075426 (2017). The authors may add a few sentences for this point as to how the Hourglass Dirac chain behaviors on this.”*

Reply: We thank the reviewer for the suggestion. As discussed in the mentioned references, the presence of drumhead-type surface states could enhance the Friedel oscillation on the surface or induce surface ferromagnetism. Since the Dirac chain metals also possess drumhead-type surface states, one expects that similar effects could appear. For a Dirac chain composed of two orthogonal loops [as in Fig.1(a) of the main text], drumhead surface states would appear on the surfaces where the two loops have finite-area projections [like (100) and (001) surfaces here], but not on the surface which is perpendicular to the two loops [(010) surface here]. Hence, interestingly, one expects that the different surfaces of a Dirac chain system could

exhibit very different behaviors in terms of the Friedel oscillation strength and surface ferromagnetism, as determined by the surface orientation relative to the chain.

In the revised manuscript, we have cited the two mentioned references and added the following discussion in the last paragraph on Page 14:

“Interestingly, the giant enhancement of the Friedel oscillation on the Be (0001) surface was also found to be due to these nontrivial surface states²⁵. In addition, with electron-electron interaction, the drumhead surface states may lead to a surface ferromagnetism, as discussed by Liu and Balents⁶¹. These effects are also expected for the Dirac chain metals, and they can be detected by surface-sensitive probes such as scanning tunneling microscopy/spectroscopy (for Friedel oscillation and superconductivity) and surface magneto-optic Kerr effect (for surface magnetism). More interestingly, the orthogonal loops dictate the presence of drumhead surface states on multiple surfaces. For the case in Fig. 1(a), the drumhead surface states would appear on (100) and (001) surfaces but not on the (010) surface. Thus, the different surfaces of a Dirac chain material could exhibit very different behaviors, e.g. in terms of the Friedel oscillation strength and the possible surface superconductivity/ferromagnetism, as determined by the surface orientation relative to the chain.”

Comment 6: *“In summary, through first-principles calculations with the symmetry analysis, the authors reported their prediction of the Hourglass Dirac chain metal in ReO₂ with a topological demonstration and its non-trivial drumhead surface states. However, I believe that all these results should be on basis of the prerequisite of the non-magnetic ReO₂. Although the early experimental results uncovered that ReO₂ is a paramagnetic metal only at ambient condition, the spin-polarized DFT (even DFT-U) calculations will be highly important to argue this point for the current manuscript at the extremely low temperatures. Therefore, I would like to ask the authors to doubly check this point, at least via some technical viable and reasonable tests. If the nonmagnetic feature of ReO₂ was theoretically and correctly demonstrated, I will strongly recommend its publication in Nature Communications because its predicted realization in specified materials for this kind of connected Dirac chain fermions.”*

Reply: We thank the reviewer for the valuable suggestions. We have fully addressed the reviewer’s concern regarding the magnetic state in the reply to Comment 1. We hope that the reviewer can now recommend the publication of our work in Nature Communications.

Reviewer #2

Comment 1: *“The manuscript by Wang et al proposed a new topological semimetal phase (the hourglass Dirac chain metal) and identified ReO₂ as a material candidate. The theoretical analyses appear to be correct and robust. I also like the fact that the proposed material candidate ReO₂ is a real compound that has been grown. For these reasons, I believe the paper is certainly valuable.”*

Reply: We thank the reviewer for the high evaluation of our work, especially for considering our analyses as “*correct and robust*” and for pointing out that “*the paper is certainly valuable*”.

Comment 2: *“On the other hand, I also think that the so-called “new” topological phase here is quite similar to the Weyl nodal chains proposed in Ref. 47. As the authors admitted, the only difference is that the degeneracy of the bands. Thus it is highly desirable that the authors can better elaborate on why the proposed Dirac chain metal is different from the Weyl chain metals in Ref. 47 in terms of the bulk states, the surface states.”*

Reply: We thank the reviewer for the helpful suggestion. In terms of the bulk states, as the reviewer pointed out, the Dirac chain has a doubled degeneracy (four-fold degenerate) compared with the Weyl chain (two-fold degenerate) proposed in [Nature 538, 75 (2016)]. Their difference is like that between Dirac semimetal and Weyl semimetal. We stress that the doubling in degeneracy is quite nontrivial: Superposing two copies of Weyl chain does not necessarily lead to a Dirac chain—they may hybridize and open a gap; the Dirac chain requires additional symmetry protection than the Weyl chain. In our case, as illustrated in Fig.3(a) in the main text, it is crucial that the two *PT*-related degenerate bands have the same glide eigenvalue, and they cross another two bands with the opposite glide eigenvalues, such that the Dirac crossing point is protected. As we mentioned, this condition fails to be valid for the $k_y = \pi$ plane, so the third Dirac loop does not appear, instead, we are only left with a pair of isolated Dirac points there.

Just like Dirac (point) semimetal being considered as a parent phase for many other topological phases (including Weyl semimetal), the Dirac chain here could also give rise to other interesting topological phases under proper symmetry breaking. For example, if time reversal symmetry is broken by a Zeeman coupling (e.g. via magnetic doping or an external magnetic field), we find that the Dirac chain may transform into a pair of Weyl loops (see Fig.R6 below). And if we break inversion symmetry, the Dirac chain may transform into a Weyl chain plus two isolated Weyl loops (see Fig.R7 below). These rich transformation properties also distinguish Dirac chain from Weyl chain.

In terms of surface states, in the Dirac chain metal here, there is a pair of spin-split drumhead surface bands for one surface (due to the broken P symmetry at surface), although there is no spin-splitting in the bulk bands due to the combined PT symmetry. This does not occur for Weyl chain metals, where both surface and bulk bands are typically without spin-degeneracy. And for Weyl chain metals, each surface typically has only one drumhead surface band.

Again we thank the reviewer for the valuable suggestion. In the revised manuscript, we have added the following discussion on Page 12:

“Compared with the Weyl chain proposed in Ref. 52, the Dirac chain here is fundamentally different due to the doubled degeneracy, similar to the distinction between Dirac and Weyl points. The added degeneracy comes from the preserved inversion symmetry, which is explicitly broken for the Weyl chain case. We stress that this doubling in degeneracy actually poses a more stringent condition regarding the symmetry protection of the band-crossings: Superposing two copies of Weyl chain does not necessarily lead to a Dirac chain—they may hybridize and open a gap; the Dirac chain requires additional symmetry protection than the Weyl chain. Indeed, if there were no such degeneracy in the current case, the (missing) loop on the $k_y = \pi$ plane would be well protected. In terms of surface states, the Dirac chain metal here possesses a pair of spin-split drumhead surface bands for a single surface, although there is no spin-splitting in the bulk bands; whereas for the Weyl chain metals, both surface and bulk bands are without spin-degeneracy, and a surface typically has only one drumhead surface band. Furthermore, like the Dirac semimetal, the Dirac chain metal may also be considered as a parent phase for other topological phases (including Weyl chain metal) under symmetry breaking. For example, by suitably breaking T or P , we can transform a Dirac chain metal into a Weyl chain metal or a nodal-loop metal (see Supplementary Information).”

And the Figs.R6 and R7 and the detailed discussion of these results are added into the Supplementary Information (Supplementary Note 6).

Figure R6 | Transformation of Dirac chain under a Zeeman field. (a,b) are results with a Zeeman field along the a -axis. The Dirac loop on the $k_z = \pi$ plane is removed, whereas the Dirac loop on the $k_x = \pi$ plane splits into two Weyl loops. (b) shows the shape of the two Weyl

loops obtained from DFT calculation (with a Zeeman energy of 0.01 eV). (c) shows the result when the Zeeman field is along the c -axis. Then the original Dirac loop on the $k_z = \pi$ plane splits into two Weyl loops.

Figure R7 | Transformation of Dirac chain under inversion symmetry breaking. (a) We break the inversion symmetry by slightly displacing two Re atoms in the unit cell along the b -axis, such that P is broken while both \tilde{M}_x and \tilde{M}_z glide mirrors are preserved. The dashed circles indicate the original locations of the two atoms. (b) shows the calculated band structure along the path from U to T. One observes that the double-degeneracy of the bulk bands is lifted. As a result, the original Dirac chain transforms into a Weyl chain plus two isolated Weyl loops, as illustrated in (c). The detailed discussion has been added in the Supplementary Information.

Comment 3: “Also, considering the fact that there have been many “new” topological semimetal phases proposed in the last 2-3 years, the authors should better explain how the proposed Dirac nodal chain metals can lead to unique possible transport and optical effects that are distinct from previously proposed topological semimetal phases.”

Reply: We thank the reviewer for the valuable suggestion. Currently, the most powerful technique for characterizing topological materials is still the angle-resolved photoemission spectroscopy (ARPES), which can directly image the bulk and surface band dispersions to compare with the calculation result. Here, we also suggest ARPES as the most direct method to distinguish the Dirac chain metal phase predicted here from other topological semimetal phases.

Besides ARPES, we also suggest the following possible effects from the nontrivial surface bands and from the bulk band-crossings that may be used to characterize Dirac chain metals. As suggested by Reviewer 1, the drumhead surface states may lead to (i) enhanced Friedel oscillations around impurities on the surface, which could be probed by scanning tunneling microscopy (STM); (ii) surface magnetism, which may be detected by the surface magneto-optic Kerr effect; or (iii) surface superconductivity, which may be probed by transport or by scanning tunneling spectroscopy (STS). These effects distinguish the Dirac chain metal from those nodal (Dirac or Weyl) point semimetals. In addition, since the presence of drumhead surface

states depends on the surface orientation, for a Dirac chain consisting of two Dirac loops as in Fig.1(a), the above-mentioned effects are possible on (100) and (001) surfaces but not on the (010) surface. This feature may help to distinguish the Dirac chain metal from the usual nodal-line semimetals with a single loop.

As for the bulk states, it was predicted that under an external magnetic field parallel to the Dirac loop plane, there will appear an almost flat Landau band at the loop energy. This will lead to a pronounced peak in DOS which can be detected by STS. For a Dirac chain as in Fig.1(a), one expects that the peak will be most pronounced when the B field is along the y -direction (parallel to both loops), and it will be relatively small when the field is not parallel to either loop. These features may distinguish Dirac chain metals from the semimetals with isolated nodal points or a single nodal loop. In addition, it has been shown that the nodal-loop dispersion leads to distinct scaling in optical absorption that $\text{Im}\epsilon(\omega)$ scales as $1/\omega$, where ϵ is the dielectric function and ω is the light frequency. This is also different from conventional metals or nodal point semimetals.

Again, we thank the reviewer for the suggestion. In the revised manuscript, we have added the following discussion on Page 13 and 14:

“..... Besides ARPES, we also suggest several interesting effects derived from the nontrivial bulk and surface states that could be useful for characterizing Dirac chain metals.

As for the bulk states, it was predicted that under an external magnetic field parallel to the Dirac loop plane, there will appear an almost flat Landau band at the loop energy⁵⁷. This will lead to a pronounced peak in the density of states which can be detected by the scanning tunneling spectroscopy. For a Dirac chain as in Fig. 1(a), one expects that the peak will be most pronounced when the B field is along the y -direction (parallel to both loops), and it will be relatively small when the field is not parallel to either loop. In addition, it has been shown that the nodal-loop dispersion leads to distinct scaling in optical absorption that $\text{Im}\epsilon(\omega)$ scales as $1/\omega$, where ϵ is the dielectric function and ω is the light frequency⁵⁸.

The drumhead-type surface states may also lead to several interesting effects. It has been argued that they could produce a huge surface density of states, which may offer a route towards high-temperature superconductivity⁶¹. The recent work by Li et al.²⁵ attributed the unusually high surface density of states on the Be (0001) surface to the drumhead surface states, which combined with the strong electron-phonon coupling found on that surface⁶⁰ may lead to a surface superconductivity (yet to be confirmed by experiment). Interestingly, the giant enhancement of the Friedel oscillation on the Be (0001) surface was also found to be due to these nontrivial surface states²⁵. In addition, with electron-electron interaction, the drumhead surface states may lead to a surface ferromagnetism, as discussed by Liu and Balents⁶¹. These effects are also expected for the Dirac chain metals, and they can be detected by surface-sensitive probes such as scanning tunneling microscopy/spectroscopy (for

Friedel oscillation and superconductivity) and surface magneto-optic Kerr effect (for surface magnetism). More interestingly, the orthogonal loops dictate the presence of drumhead surface states on multiple surfaces. For the case in Fig. 1(a), the drumhead surface states would appear on (100) and (001) surfaces but not on the (010) surface. Thus, the different surfaces of a Dirac chain material could exhibit very different behaviors, e.g. in terms of the Friedel oscillation strength and the possible surface superconductivity/ferromagnetism, as determined by the surface orientation relative to the chain.”

REVIEWERS' COMMENTS:

Reviewer #1 (Remarks to the Author):

The authors already emphasized all points raised in my last report and I would like to support its publication in this version.

Reviewer #2 (Remarks to the Author):

I have read the replies from the authors to the whole review report. I think the authors have carefully and correctly addressed the questions. I therefore recommend the paper for publication .

Response to Reviewer #1

Comment: *“The authors already emphasized all points raised in my last report and I would like to support its publication in this version.”*

Response: We thank the reviewer for the many valuable suggestions in the previous report and for supporting the publication of our work in the current version.

Response to Reviewer #2

Comment: *“I have read the replies from the authors to the whole review report. I think the authors have carefully and correctly addressed the questions. I therefore recommend the paper for publication.”*

Response: We thank the reviewer for the careful evaluation of our work and for recommending its publication in Nature Communications.